# Efficacy of Pembrolizumab in Advanced Melanoma: A Narrative Review

**DOI:** 10.3390/ijms241512383

**Published:** 2023-08-03

**Authors:** Giulio Rizzetto, Edoardo De Simoni, Elisa Molinelli, Annamaria Offidani, Oriana Simonetti

**Affiliations:** Clinic of Dermatology, Department of Clinical and Molecular Sciences, Polytechnic University of Marche, 60126 Ancona, Italy; g.rizzetto@pm.univpm.it (G.R.); edodesimoni@hotmail.it (E.D.S.); molinelli.elisa@gmail.com (E.M.); a.offidani@ospedaliriuniti.marche.it (A.O.)

**Keywords:** advanced melanoma, pembrolizumab, immunotherapy

## Abstract

Pembrolizumab has been shown to increase survival in patients with metastatic melanoma. Considering the numerous oncoming studies, we decided to conduct a narrative review of the latest efficacy evidence regarding the use of pembrolizumab, alone or in combination, in patients with metastatic melanoma. A search was conducted in PubMed using “pembrolizumab,” and “metastatic melanoma” as keywords, considering studies from 2022 onward. We reviewed pembrolizumab and associations, cost-effectiveness, virus, advanced acral melanoma, long-term outcomes, real-life data, biomarkers, obesity, and vaccines. In conclusion, pembrolizumab is a fundamental option in the therapy of metastatic melanoma. However, a certain group of patients do not respond and, therefore, new combination options need to be evaluated. In particular, the use of vaccines tailored to tumor epitopes could represent a breakthrough in the treatment of resistant forms. Further studies with larger sample numbers are needed to confirm the preliminary results.

## 1. Introduction

Melanoma is one of the most problematic malignancies for public health since melanoma is the most life-threatening skin cancer and is associated with 75 per cent of skin cancer deaths. The progressive increase in new diagnoses appears to be due to improvements in surveillance and early diagnosis of melanoma, although sun exposure and sunbeds also seem to play an important role. In addition, about 20% of patients may develop an advanced (unresectable/metastatic) form [1]. However, the possibility of evaluating new markers for lymph node involvement [2,3] and new therapeutic regimens for metastatic melanoma allowed a substantial reduction in mortality [4]. Immunologic checkpoint inhibitors (ICIs), particularly ipilimumab, pembrolizumab, and nivolumab, permitted a revolution of the approach to the metastatic patient, inducing a relevant increase in survival [5,6,7]. Ipilimumab, an inhibitor of Cytotoxic T-lymphocyte-associated protein-4 (CTLA-4), in monotherapy, allowed an increase in the 10-year survival of metastatic patients from 10% to 22% [8].

The programmed death (PD-1) inhibitors nivolumab and pembrolizumab were also shown to increase survival in metastatic patients [9,10,11,12,13]. In three clinical trials in patients with advanced melanoma (KEYNOTE-001, KEYNOTE-002, and KEYNOTE-006), therapy with pembrolizumab alone resulted in a 30–40% complete clinical response [9,11,14,15,16,17]. Specifically, in the KEYNOTE-001 study on advanced melanoma, pembrolizumab monotherapy resulted in a 3-year overall survival (OS) of 41% in patients previously treated with ipilimumab and 45% in therapy-naïve patients [13]. New studies are currently emerging in the literature regarding the efficacy of pembrolizumab in patients with metastatic melanoma, even considering the long-term efficacy in real life.

Pembrolizumab is a humanized IgG4/kappa anti-PD1 monoclonal antibody produced in hamster ovarian cells. The PD-1 receptor is a negative regulator of T-cell activity that has been shown to be involved in the control of T-cell immune responses. KEYTRUDA potentiates T-cell responses, including antitumor responses, by blocking PD-1 binding to PD-L1 and PD-L2, which are expressed on antigen-presenting cells and may be expressed by tumors or other cells in the tumor microenvironment [18]. According to current indications, pembrolizumab is indicated as monotherapy in the treatment of adults and adolescents > 12 years of age with unresectable or metastatic melanoma [18]. In addition, pembrolizumab as monotherapy is indicated in the adjuvant treatment of adults and adolescents > 12 years old with melanoma stage IIB, IIC, or III and who had complete tumor resection [18]. Recently, the benefit of pembrolizumab as neoadjuvant therapy in stage III and IV melanoma also emerged, with a longer event-free survival in patients using pembrolizumab as neoadjuvant and adjuvant compared to the pembrolizumab adjuvant group [19].

A recent meta-analysis [20] compared the use of pembrolizumab with conventional chemotherapy drugs for the treatment of advanced forms of melanoma, showing promising results for pembrolizumab (pembrolizumab monotherapy group 2 mg/kg with OS 13.4 months, 10 mg/kg group OS 14.7 months, chemotherapy group OS 11.0 months). However, there were insufficient data to confirm the statistical significance.

In cutaneous melanoma, in addition to tumor mutational burden (TMB), MHC-II expression, plodia, heterogeneity, and tumor purity were also associated with a response to pembrolizumab. Specifically, MHC-II expression on tumor cells was found to be predictive of response to anti-PD1, and it was hypothesized that they represented a subset of tumors capable of stimulating CD4 T-cells or CD8 T-cells. In support of this, MHC-II transcriptomics was correlated with the expression of CD4, PRF1, and GZMA (cytolytic molecules) [21].

Considering the numerous studies being published, we decided to conduct a narrative review of the literature reporting the latest efficacy evidence regarding the use of pembrolizumab, alone or in combination, in patients with metastatic melanoma.

## 2. Materials and Methods

A search was conducted in PubMed using “pembrolizumab,” and “metastatic melanoma” as keywords. Only English language studies were included in the narrative review. Studies from 2022 onward were considered as a time limit. We identified a total of 162 studies. Only studies, including clinical trials, observational studies, or metanalyses, reporting efficacy data about pembrolizumab, alone or in combination or in specific conditions, were included. Studies on safety or adverse effects were not included in this narrative review, as well as case reports or case series. We decided to include a total of 26 studies that met the inclusion criteria. We reported studies, even with preliminary data from clinical studies, that were published from 2022 onwards, while ongoing clinical trials with no published data were not included in this review. The purpose is to summarize the latest evidence produced and reviewed regarding the efficacy of pembrolizumab in patients with metastatic melanoma since, to the best of our knowledge, a monothematic and specific overview of pembrolizumab in the most recent studies is not currently available in the literature.

## 3. Results

From our review of the literature in PubMed, a total of 143 articles were found. After excluding the studies that did not meet the search criteria, a total of 20 studies were reviewed.

### 3.1. Pembrolizumab and Associations

In a multicenter, double-blind phase III study, the combination of pembrolizumab and talimogene laherparepvec (T-VEC) versus placebo and pembrolizumab was evaluated in 692 patients with metastatic/unresectable melanoma who were naive for anti-PD1. In both groups, pembrolizumab was administered at 200 mg once every 3 weeks. T-VEC and pembrolizumab did not significantly improve the progression-free survival (PFS) and OS compared with the placebo/pembrolizumab group. Thus, in this study, it appears that combination therapy does not result in improvement regarding the parameters considered [22].

A recent phase I study evaluated the combination of pembrolizumab, vemurafenib, and cobimetinib as a first line in BRAF V600E/K mutated BRAF metastatic patients. There were two groups treated with pembrolizumab/vemurafenib and pembrolizumab/vemurafenib/cobimetinib, respectively. The median OS was 23.8 months for vemurafenib/pembrolizumab, while the triple combination did not allow the calculation of the OS as it was discontinued due to the occurrence of side effects, leading to the closure of the study [23]. BRAF inhibitors (BRAFi) appear to be able to modulate the tumor microenvironment (TME). In fact, on the one hand, the administration of a BRAFi in a BRAF-mutant melanoma model resulted in increased expression of CD40 ligand and interferon-gamma on tumor-infiltrating CD4+ lymphocytes (TILs) and decreased T-reg and myeloid cells on the other hand, suggesting antitumor modulation of the TME [23]. Furthermore, an increase in the T-cell exhaustion markers TIM-3 and PD1 was correlated with BRAFi in patients with metastatic melanoma [23]. However, the triple therapy proposed in this study did not allow the progression of the trial due to significant adverse events, such as dermatitis (*n* = 8), hepatitis (*n* = 1), arthralgias (*n* = 1), and QTc prolongation (*n* = 1).

Despite the failure of previous combinations, a phase Ib/II study of 24 patients with metastatic melanoma evaluated the efficacy of pembrolizumab and all-trans retinoic acid (ATRA), achieving an overall response rate of 71%, of which 50% were in complete response, and a 1-year OS was 80%. The aim of this study was to target circulating myeloid-derived suppressor cells (MDSCs), seeking to enhance the action of immunotherapy. The promising results open the way for further studies and represent an interesting new therapeutic option [24]. MDSCs are cells characterized by high immune function. Since MDSCs are immature cells, they can change following exposure to agents such as ATRA, a derivative of vitamin A. Specifically, ATRA appears to reduce both the frequency and functions of MDSCs through the activation of ERK1/2, the upregulation of glutathione synthase, and increased glutathione. The latter reduces ROS production and, consequently, results in the terminal differentiation of MDSCs. Thus, it is hypothesized that combining pembrolizumab with ATRA could reduce the frequency of MDSCs and improve the efficacy of pembrolizumab, particularly by reducing polymorphonucleate-MDSCs [24].

Finally, in a phase Ib study of patients with metastatic melanoma, the association between pembrolizumab (200 mg IV every 3 weeks) and escalating doses of IL-2 (6000 or 60,000 or 600,000 IU/kg IV bolus every 8 h up to 14 doses per cycle) was evaluated in cohorts of three patients. Nine patients were included, and a partial response was obtained in one (11%), previously treated with pembrolizumab, with high doses of IL-2. Another patient showed a stable condition after 4.5 years of follow-up, even though he had not had pembrolizumab treatment prior to the study. None of the patients showed dose-limiting toxicities. For this reason, treatment with pembrolizumab and IL-2 could be conducted, especially in those patients who did not respond to pembrolizumab therapy alone. However, an evaluation on a larger sample is needed [25] (Table 1).

### 3.2. Pembrolizumab Cost-Effectiveness

A recent study based on a partitioned survival model with a 1-week cycle length and a 20-year base-case time horizon evaluated the cost-effectiveness of pembrolizumab versus paclitaxel and carboplatin as a first line in patients with metastatic or unresectable melanoma. Considering that an increase of 2.63 life-years and an increase of 2.24 quality-adjusted life-years was estimated, pembrolizumab emerges as an effective treatment option in metastatic patients as a first line and is also cost-effective, not exceeding the total cost of treatment the three times the per capita gross domestic Chinese [26]. This confirms that first-line therapy with pembrolizumab is not only an effective option but also an economically viable option in countries with a per capita product similar to China.

**Table 1 ijms-24-12383-t001:** Efficacy of pembrolizumab in association: summary of reviewed studies on metastatic/unresectable melanoma.

Study	Type of Study	Treatments	Results
Chesney et al., 2023 [22]	double-blind phase III study	P/T-VEC vs. P/placebo	T-VEC/pembrolizumab not significantly improve PFS and OS compared with placebo/pembrolizumab
	692 patients anti-PD1 naive		
Shaikh et al., 2022 [23]	phase I study	P/vemurafenib vs. P/vemurafenib/cobimetinib	median OS was 23.8 months P/vemurafenib P/vemurafenib/cobimetinib discontinued for sides
	BRAF V600E/K metastatic patients		
Tobin et al., 2023 [24]	phase Ib/II study 24 patients	P/ATRA	overall response rate 71%, of which 50% complete response 1-year OS was 80%.
Silk et al., 2023 [25]	phase Ib study 9 patients	P/ escalating doses of IL-2 (6000 or 60,000 or 600,000 IU/kg IV bolus every 8 h up to 14 doses per cycle)	partial response 1 (11%), with high doses IL-2 1 patient stable condition after 4.5 years of FU No dose-limiting toxicities. Evaluation of a larger sample is needed
Silk et al., 2023 [27]	phase 1b study 36 patients	Coxsackievirus A21 (V937) intratumorally + P	objective response in 47%, complete response in 22% of cases

Abbreviations: talimogene laherparepvec (T-VEC), progression-free survival (PFS), overall survival (OS), pembrolizumab (P), all-trans retinoic acid (ATRA), recurrence-free survival (RFS).

### 3.3. Pembrolizumab and Virus

Coxsackievirus A21 (V937) is an oncolytic virus that has been shown to be effective against metastatic melanoma. Oncolytic viruses are believed to act against tumors through two distinct mechanisms: direct lysis of tumor cells and the activation of a specific antitumor immune response. Coxsackievirus is an RNA virus, which can cause mild upper respiratory symptoms, and V937 is a selected, genetically unmodified, oncolytic strain that can enter cells by binding to intracellular adhesion molecule-1 (ICAM-1) and decay-accelerating factor (DAF) receptors. Both of these receptors are specifically expressed on the surface of melanoma cells. From a pharmacodynamic point of view, oncolytic viruses result in increased production of interferon, CD8+ T-cells and the expression of programmed death ligand 1 (PD-L1) in the tumor microenvironment, as well as reduced populations of suppressor T-cells. For this reason, it was assumed that they could improve the responses when combined with inhibitors inhibiting PD-1/PD1-L or cytotoxic T lymphocyte antigen 4 (CTLA-4) [27].

In a recent phase 1b study of 36 patients, Coxsackievirus A21 (V937) was administered intratumorally in combination with pembrolizumab, achieving an objective response in 47% of cases and a complete response in 22% of cases. Interestingly, the density of the intratumoral CD3+CD8− T-cells was lower in the responders, suggesting that the combination of Coxsackievirus A21 (V937) and pembrolizumab might help overcome the limitations of a nonimmunologically “active” tumor environment [27]. Surprisingly, the responses were not correlated with an increase in viral signaling proteins (RIG-I, TLR7, and TLR8), viral entry proteins (ICAM-1 and DAF), or PD-L1, nor was an inflamed tumor microenvironment found. In fact, the levels of CD3+CD8− infiltrating T lymphocytes in the pre-treatment tumor samples were actually lower in the responders than in the non-responders. This phenomenon could hypothetically be explained by considering that the pre-treatment tumor samples of the responders showed a lower number of regulatory T-cells than the non-responders [27]. Another hypothesis that may explain how fewer tumor-infiltrating lymphocytes may favor a better response is that fewer immune cells were immediately available in the tumor to fight the Coxsackievirus infection, resulting in more robust viral replication and an increased circulating viral load. A significant increase in IP-10/CXCL10 and a smaller increase in MDC/CCL22 were also observed in the serum of patients. CXCL10 plays an important role in the recruitment of antitumor T-cells in melanoma. Furthermore, CXCL10 via the CXCR3 receptor promotes the migration of lymphocytes to dendritic cells, which is necessary for the response to PD-1 blockade [28]. Clinically, it has been shown that elevated pre-treatment CXCL9 and CXCL10 levels correlate with the response to anti-PD-L 1 therapy in patients with non-small cell lung cancer, and CXCL9 and CXCL10 increase in the first months of treatment in melanoma patients responding to PD-1 inhibitor therapy. Finally, elevated pre-treatment CXCL9 and CXCL10 levels appear to correlate with the response to anti-PD1 therapy in patients with non-small cell lung cancer [29]. Similarly, we found an increase in CXCL9 and CXCL10 in the first months of treatment with PD-1 inhibitors in responders with melanoma.

In light of these data, it is hypothesized that V937 infection increases the production of interferon-gamma and CXCL10 by immune cells in the tumor microenvironment, particularly macrophages, which promote the response to anti-PD1 by properly guiding activated lymphocytes to dendritic cells [27]. In our opinion, the use of oncolytic viruses could be a promising solution, helping to overcome tumor immune escape mechanisms.

In a pilot study, the efficacy of intratumorally injected ONCOS-102, an oncolytic adenovirus expressing human GM-CSF, was evaluated in combination with pembrolizumab. In total, 21 patients with advanced melanoma with disease progression after treatment with pembrolizumab were enrolled. The treatment was well-tolerated, with few adverse effects (pyrexia, nausea, chills) and no dose-limiting toxicities, being objectively effective at reducing the size of one or more non-injected metastases, highlighting the systemic action of the treatment. Serial analyses were also performed on tumors injected with ONCOS-102 using RNA expression and immunofluorescence, showing the correlation between the immune cell level and persistence and patient outcome [30].

### 3.4. Pembrolizumab and Advanced Acral Melanoma

A retrospective cohort study evaluated the efficacy in 325 patients with unresectable acral melanoma of PD-1 inhibitors, including pembrolizumab, in combination or not with ipilimumab. The groups were treated with anti-PD1 (184, 57%), anti-PD1/ipilimumab (59, 18%), and ipilimumab alone (82, 25%). The objective response rate (ORR) was significantly higher in the anti-PD1/ipilumumab combination group (43%) than in anti-PD1 alone (26%) and ipilimumab alone (15%), respectively. However, anti-PD1/ipilimumab did not lead to a significant difference in the OS and PFS compared with anti-PD1 alone [31]. For this reason, further trials considering the acral melanoma subgroup are needed to evaluate the best treatment approach.

### 3.5. Pembrolizumab, Long-Term Outcomes, and Real-Life Data

Regarding long-term outcomes, 5-year OS data for pembrolizumab in metastatic melanoma have been published, with reference to the KEYNOTE-001 studies [32] and KEYNOTE-006 [33]. The results were comparable to those of nivolumab, showing a 5-year OS of 41% and 43%, respectively, in the two trials [34].

In a study of 103 Chinese patients with metastatic melanoma, a 3-year follow-up was performed after failure of the first treatment and second treatment with pembrolizumab 2 mg/kg every 3 weeks for ≤35 cycles. The median OS was 13.2 months, and the 36-month OS rate was 22.3%. It is interesting that the median OS data by melanoma subtype are reported, specifically 14.8 months for acral, with a significant difference between acral PD-L1-positive disease (median OS 22.8 months) and acral PD-L1-negative disease (8.4 months). In addition, pembrolizumab therapy as a second-line therapy was shown to have good tolerability [35]. These studies suggest that a good therapeutic response is maintained even in long-term follow-up, partly due to the tolerability of the treatment.

In a retrospective study on 132 patients with metastatic melanoma treated with ICIs, including pembrolizumab, ref. [36] Perez et al. report that 34.8% achieved a complete clinical response, while the PFS was 97.5% at 1 year and 94.7% at 3 years after treatment discontinuation. Thus, this real-life study suggests that the discontinuation of ICIs in those metastatic patients with complete remission of the disease is feasible, keeping the clinical condition stable at 3 years. However, it is also necessary to evaluate these aspects in studies with a larger sample size.

In another national retrospective study of 5097 New Zealand patients with metastatic melanoma treated with ICIs, including pembrolizumab, a 1-year OS of 72% was reported, while the 2-year OS was 60%, confirming the data from previous clinical trials [37].

Finally, a recent study of 1037 patients with metastatic melanoma compared the efficacy of nivolumab and pembrolizumab. Both have approval for the treatment of metastatic melanoma, but the choice between the two is often left to the clinician. The results of this study showed that the median OS was 17.4 months for pembrolizumab and 20 months for nivolumab, while the estimated OS at 2 and 3 years was 42/34% for pembrolizumab and 47/37% for nivolumab. The differences in the OS, PFS, and overall response were not statistically significant in naive patients with metastatic melanoma, confirming the clinician’s role in the treatment choice [38].

### 3.6. Pembrolizumab and Biomarkers

Possible new markers of therapeutic response to pembrolizumab recently emerged in a study of 46 metastatic melanoma patients. CD103 expressed on CD8+ T lymphocytes was associated with a complete response to pembrolizumab with statistical significance when the density (*p* = 0.04) and proportions (*p* = 0.012) of this T-cell population increased. Improved survival correlated with the increased proportion of CD8+ CD103+ T-cells (*p* = 0.0085) and also with the reduced expression of periplakin (*p* = 0.012) and periplakin + SOX10 (*p* = 0.0012) [39]. This preliminary study highlights that the role of certain cellular markers could help select those metastatic patients most promising to respond to pembrolizumab monotherapy; however, further studies on a larger sample are needed.

In a study of 68 metastatic melanoma patients, tryptophan and glucose metabolism were evaluated using C11-labeled α-methyl tryptophan (C11-AMT) and fluorodeoxyglucose (FDG) PET imaging of tumor lesions. In addition, the expression of tryptophan-metabolizing enzymes (TMEs; TPH1, TPH2, TDO2, IDO1) and the tryptophan transporter LAT1 were evaluated using immunohistochemistry. In this study, it was found that elevated tryptophan metabolism in metastatic melanoma is a predictor of a poor response to pembrolizumab [40]. In fact, melanoma cancer cells are presumed to possess a higher tryptophan uptake capacity than TILs. This type of dysregulation of tryptophan metabolism is believed to be responsible for the reduced proliferation of TILs due to the reduced availability of this essential amino acid. Furthermore, TILs switch from an active to a ‘dormant’ state. On the other hand, the metabolism of tryptophan in tumor cells results in increased local levels of serotonin and kynurenines, which act by negatively regulating TILs [40].

Interestingly, some soluble biomarkers have also been proposed. In a recent study of 41 patients with metastatic melanoma, the plasma levels of soluble PD-1 (sPD-1), soluble programmed cell death ligand 1 (sPD-L1), butyrophilins (BTNs) 2A1, 3A1, and body mass index (BMI) were evaluated as predictors of the efficacy of anti-PD1 therapy. Low levels of sPD-1 plus high levels of sBTN2A1 were associated with a better overall response rate. In addition, patients with a BMI ≥ 25 and sPD-1 < 11.24 ng/mL had a longer time to treatment failure (*p* < 0.001) [41].

Regarding the imaging technique, another study used positron emission tomography (PET) with zirconium-89 (89Zr)-labeled pembrolizumab prior to pembrolizumab treatment to evaluate the predictive ability of the response to anti-PD1. In 11 patients with metastatic melanoma, 89Zr-pembrolizumab tumor uptake was calculated, and a statistically significant direct correlation with the clinical response, OS, and PFS was obtained. Although this method is interesting in the landscape of pembrolizumab efficacy markers, the small sample size requires further studies to confirm its clinical utility [42] (Table 2).

Texture analysis features at contrast medium CT of melanoma metastases also appear to be a predictor of the response to treatment with anti-PD1. In a study of 127 metastases, three items were identified as independent predictors of response to therapy: skewness at the fine texture scale, skewness at the medium texture scale, and variation of entropy at the fine texture scale. Although further studies are needed to confirm their validity, CT methods may also be an aid in assessing the response to anti-PD1 immunotherapy [43].

### 3.7. Pembrolizumab and Vaccines

The phase II KEYNOTE-D36 clinical trial will evaluate the efficacy and safety of the combination of pembrolizumab and EVX-01, a personalized neoepitope vaccine, in patients with metastatic or unresectable melanoma. EVX-01 consists of multiple peptides that represent a neoepitope expressed only in the patient’s specific tumor, allowing a tumor-specific immune response [44].

Regarding the use of vaccines in combination with pembrolizumab, data were recently presented from a KEYNOTE-942 clinical trial in patients with completely resected, high-risk melanoma. The patients were treated with mRNA-4157/pembrolizumab (107) or pembrolizumab alone (50). Nine doses of mRNA-4157 (1 mg) were administered every 3 weeks, while pembrolizumab 200 mg was administered intravenously for 18 cycles. After 18 months, the recurrence-free survival (RFS) was 78.6% in the combination group versus 62.2% in the group with pembrolizumab alone. In addition, the combination group showed a reduction in the risk of recurrence or death by 44% (HR = 0.561; 95% CI: (0.309, 1.017). These results are very promising, highlighting the therapeutic possibilities of the combination of an adjuvant standard of care and a vaccine tailored to specific tumor neoantigens; however, a phase III study with a larger sample size is needed to confirm the results [45,46]. In particular, this mRNA vaccine encodes up to 34 tumor neoantigens and is produced based on the specific mutation sequence of the patient’s tumor. Once administered into the body, the neoantigen sequences encoded by the RNA are translated endogenously and subjected to natural antigen processing and presentation by antigen-presenting cells, a crucial step in adaptive immunity.

## 4. Discussion

Current therapeutic options in advanced melanoma include PD-1 inhibitors associated or not with CTLA-4 or lymphocyte-activating gene (LAG)-3 inhibitors and in cases of BRAF V600E/K mutated BRAF inhibitors, called targeted therapy [11,47]. However, the DREAMseq study showed that immunotherapy followed by target therapy in case of progression increased the OS by 20% compared with reversing the two treatments [48]. This is because BRAF inhibitors are supposed to modulate the tumor immune microenvironment by increasing interferon-gamma production on intratumoral CD4+ T lymphocytes and increasing CD8+ TILs [49,50].

Currently, the guidelines suggest the use of ICIs with anti-PD1, in association or not with anti-CTLA4, in patients with metastatic melanoma, regardless of BRAF status, allowing long-lasting responses and long-time survival in the responders [51,52].

Specifically, clinical trials of ICIs in patients with metastatic melanoma report a 5-year OS of up to 60% and a 5-year ongoing response rate of up to 62% [49,53]. However, about 40–60% of treated patients do not respond to ICIs, and about 30% of treated patients relapse in the following 2 years [7,11,32,53,54,55].

Therefore, the use of vaccines in combination with pembrolizumab could be a solution for all those patients who do not respond to immunotherapy due to immune escape. Clinical trial data show that 50 to 64% of melanoma patients, even if on immunotherapy, will experience disease progression after 1 to 5 years [47]. The mechanisms of resistance may be related to several aspects. The reduced expression of molecular targets, particularly PD-L1, on neoplastic cells results in primary tumor resistance to pembrolizumab [56,57] 

Other causes of resistance to immunotherapy are a low neoantigen load on tumor cells, allowing them to overcome immune surveillance [58], and immunosuppression brought about by the tumor microenvironment [59,60,61,62].

Another aspect to be considered is the use of markers that could help predict the response to pembrolizumab, thus allowing the selection of patients who might benefit from association with other treatments, such as personalized incoming vaccinations. In our opinion, the use of texture analysis features to contrast medium CT of melanoma metastases could be a method that could be easily integrated into the staging of the metastatic patient; however, further studies are needed to confirm its usefulness [40].

Furthermore, an evaluation of the tumor microenvironment appears to be useful in assessing the prognosis of melanoma [63,64,65,66,67] and provides insight into the efficacy of pembrolizumab therapy [36], although the data need further confirmation on larger samples. For example, although obesity is a major risk factor for cancer mortality [68], an “obesity paradox” emerges as obesity correlates with better clinical outcomes in patients with metastatic melanoma, especially when treated with anti-PD1. In a retrospective study of 266 patients with metastatic melanoma treated with pembrolizumab or nivolumab, it emerged that obesity had a protective role. Specifically, a 10 cm^2^/m^2^ increase in the visceral fat index (VFI) was associated with longer overall survival after adjusting for covariates, but this was not maintained when the systemic immune-inflammation index was adjusted for. Thus, the prognostic role of visceral obesity is conditional on the inflammatory status. Obesity appears to determine an increase in PD-1 expression, induces T-cells to release more PD-1 protein, and increases the secretion of adiponectin and leptin from adipose tissue. These alterations result in increased T-cell depletion and dysfunction, which promote tumor progression. On the other hand, this provides a better understanding of the relationship between obesity and ICIs, as these agents remove the inhibitory signals of T-cell activation and promote an effective antitumor response [69]. On the other hand, another real-life study on 1070 patients reported that obese or overweight patients had a higher risk of immune-related adverse events, showing that these patients, on the one hand, respond better clinically than normal-weight patients but at the same time are immunologically more reactive, experiencing more cutaneous, gastro-intestinal, hepatic, pulmonary, rheumatic, and endocrine immune-adverse events [70].

It is important to note that therapy with pembrolizumab can be combined with BRAF inhibitors. The BRAF V600 mutation is found in approximately 35–50% of metastatic melanomas and is associated with increased disease aggressiveness and reduced survival [71,72]. A recent multicenter, double-blind phase III trial (STARBOARD) of 624 patients with BRAF V600E/K metastatic melanoma compared the efficacy of triple therapy (encorafenib, binimetinib, and pembrolizumab) versus placebo plus PD-1 monotherapy (pembrolizumab). However, the data from this study are not currently available.

Another interesting association is with pembrolizumab and metformin, which seems to be able to increase the efficacy of pembrolizumab by reducing CD8+ T-cell apoptosis, reducing intratumoral hypoxia and reducing PD-L1 expression, thus favoring a better antitumor cell response [73]. All this is an example of how drug repurposing can be a useful strategy to improve the efficacy of current treatments, especially in resistant cases. In fact, even a molecule such as cetirizine can be combined with anti-PD1 to improve its efficacy. In a retrospective study [74] on patients with stage IIIb-IV melanoma, the cetirizine/anti-PD1 combination, including pembrolizumab, showed a better PFS and OS than treatment without cetirizine. This may be explained by the polarization of M1 macrophages induced by cetirizine, enhancing the anti-PD1 immune response.

Finally, a recent meta-analysis by Li et al. [75] evaluated the efficacy and safety of ICIs in 10,090 patients from randomized clinical trials. These data show that pembrolizumab at a dosage of 10 mg/kg is the treatment option with better efficacy and safety, comparable with nivolumab, among ICIs in monotherapy. However, the best OS and PFS are obtained by combining nivolumab 1 mg/kg and ipilimumab 3 mg/kg.

## 5. Conclusions

In conclusion, pembrolizumab represents a fundamental option in the therapy of metastatic melanoma, as confirmed by data in real life and in long-term follow-up. However, a certain proportion of patients do not respond to this therapy and, therefore, new combination options need to be evaluated. Among the new treatments, the use of vaccinations tailored to tumor epitopes could represent a breakthrough in the treatment of resistant forms, but further studies with larger sample numbers are needed to confirm the preliminary results. Biomarkers that may allow the selection of patients with a better chance of a response to pembrolizumab therapy are also very important. In particular, the evaluation of CD103 expression and the CT characteristics of metastases could in the future be easily integrated into diagnostic practices, allowing a more specific therapeutic choice. However, further studies with a larger sample are required to confirm the validity of these biomarkers.

## Figures and Tables

**Table 2 ijms-24-12383-t002:** Pembrolizumab and biomarkers: summary of reviewed studies on metastatic/unresectable melanoma.

Study	Type of Study	Biomarkers	Results
Edmonds et al., 2022 [39]	preliminary study 46 patients	CD103 Periplakin periplakin + SOX10	increased density/proportions CD103 CD8+ T associated with complete response to P Reduced expression of periplakin and periplakin + SOX10 associated with improved survival during P
Oldan et al., 2023 [40]	Prospective clinical trial 68 patients	C11-AMT FDG PET tryptophan-metabolizing enzymes	elevated tryptophan metabolism predictor of poor response to P.
Incorvaia et al., 2023 [41]	Prospective cohort study 41 patients	sPD-1 sPD-L1 BTNs 2A1, 3A1	Low levels of sPD-1 + high levels of sBTN2A1 associated with a better overall response rate. patients with BMI ≥ 25 and sPD-1 < 11.24 ng/mL had longer time to treatment failure.
Kok et al., 2022 [42]	Prospective study 11 patients	(89Zr)-labeled P PET	89Zr-P tumor uptake significant direct correlate with clinical response, OS, and PFS
Bonnin et al., 2022 [43]	Retrospective study 127 metastases	Texture analysis features at contrast medium CT of melanoma metastases	3 items predictors of favorable response to P skewness at FTS, skewness at MTS, variation of entropy at FTS

Abbreviations: pembrolizumab (P), C11-labeled α-methyl tryptophan (C11-AMT), fluorodeoxyglucose (FDG), tryptophan-metabolizing enzymes (TMEs; TPH1, TPH2, TDO2, IDO1), soluble PD-1 (sPD-1), soluble programmed cell death ligand 1 (sPD-L1), butyrophilins (BTNs), zirconium-89 (89Zr), progression-free survival (PFS), overall survival (OS), fine texture scale (FTS), medium texture scale.

## Data Availability

Not applicable.

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
