# Peer review of "Efficacy of Pembrolizumab in Advanced Melanoma: A Narrative Review"

_ijms, 2023, doi:10.3390/ijms241512383_

Round 1

Reviewer 1 Report

1. "P" should be capital in pembrolizumab in the abstract.

2. I would suggest to remove the subheadings like "Introduction", "Methods", etc. on the abstract, as it breaks the flow of the reader. 

3. "C" in considering should be small as the sentence didn't finished at melanoma.

4. Please provide the references for the following statement:

"According to current indications, pembrolizumab is indicated as monotherapy in the treatment of adults and adolescents > 12 years of age with unresectable or metastatic melanoma."

5. In line 60, correct the spelling of mutational.

6. Please extend the Materials and Methods section by providing different parameters, such as, how many studies you have found using your criteria? How many have you used and rationale behind eliminating others?

7. line 90, correct spelling of combination.

8. Please provide the full form of OS at its first appearance in the text.

9. Expand vit. A in line 113.

10. Please maintain the uniformity for the drug 'Pembrolizumab', somewhere "P" is capital and rest its small.

11. Please re-write the followinf sentence:

One hypothesis that might explain this is that the pre-treatment tumour samples of responders include fewer regulatory cells 

12. At line 177, please correct anti-PD-L1.

13. What is the point of providing two lines as paragraph at line 188 and 189.

14. The following sentence is grammatically incoorect.

"Included were anti-PD-1 184 (57%), anti-PD-1/ipilimumab 59 (18%), and 82 (25%) ipilimumab alone."

15. Please omit extra "e" at line 212.

16. Line 217, its should be months not month, without hyphen.

17. Please expand Discussion and conclusions section.

1. A thorough check for the grammatical errors is needed.

2. I would suggest to lookout for the punctuation errors throughout the manuscript.

Author Response

  1. "P" should be capital in pembrolizumab in the abstract.

done

  1. I would suggest to remove the subheadings like "Introduction", "Methods", etc. on the abstract, as it breaks the flow of the reader. 

done

  1. "C" in considering should be small as the sentence didn't finished at melanoma.

We added a mark before.

  1. Please provide the references for the following statement:

"According to current indications, pembrolizumab is indicated as monotherapy in the treatment of adults and adolescents > 12 years of age with unresectable or metastatic melanoma."

Done, [18]

  1. In line 60, correct the spelling of mutational.

Done, thank you

  1. Please extend the Materials and Methods section by providing different parameters, such as, how many studies you have found using your criteria? How many have you used and rationale behind eliminating others?

Done

A search was conducted on Pubmed using "pembrolizumab," and "metastatic melanoma" as keywords. Only English language studies were included in the narrative review. Studies from 2022 onward were considered, as a time limit. We identified a total of 162 studies. Only studies, clinical trials or observational, reporting efficacy data about pembrolizumab, alone or in combination or in specific conditions, were included. Studies on safety or adverse effects were not included in this narrative review, as well as case-report or case-series. We decided to include a total of 26 studies that met the inclusion criteria. We reported studies, even with preliminary data, from clinical studies that were published from 2022 onwards, while ongoing clinical trials with no published data were not included in this review. The purpose is to summarize the latest evidence produced and reviewed regarding the efficacy of pembrolizumab in the patients with metastatic melanoma, since, at the best of our knowledge, a monothematic and specific overview about pembrolizumab on the most recent studies is not currently available in the literature.

  1. line 90, correct spelling of combination.

Done, the combination

  1. Please provide the full form of OS at its first appearance in the text.

Done, line 39

Specifically, in the KEYNOTE-001 study on advanced melanoma, pembrolizumab monotherapy resulted in a 3-year overall survival (OS)

  1. Expand vit. A in line 113.

Done, Vitamin A

  1. Please maintain the uniformity for the drug 'Pembrolizumab', somewhere "P" is capital and rest its small.

Done

  1. Please re-write the following sentence:

One hypothesis that might explain this is that the pre-treatment tumour samples of responders include fewer regulatory cells 

Done

This phenomenon could hypothetically be explained by considering that pre-treatment tumour samples of responders showed a lower number of regulatory T-cells than non responders.

  1. At line 177, please correct anti-PD-L1.

Done

  1. What is the point of providing two lines as paragraph at line 188 and 189.

Corrected, we eliminated the paragraph

  1. The following sentence is grammatically incoorect.

"Included were anti-PD-1 184 (57%), anti-PD-1/ipilimumab 59 (18%), and 82 (25%) ipilimumab alone."

Done.

The groups were treated with anti-PD-1 (184, 7%), anti-PD-1/ipilimumab (59, 18%), and ipilimumab alone (82, 25%).

  1. Please omit extra "e" at line 212.

Done

  1. Line 217, its should be months not month, without hyphen.

done

  1. Please expand Discussion and conclusions section.

Done

Furthermore, evaluation of the tumor microenvironment, appears to be useful in assessing the prognosis of melanoma [64-68], and to be able to give insight into the efficacy of pembrolizumab therapy [35], although the data need further confirmation on larger samples. For example, although obesity is a major risk factor for cancer mortality [43], an "obesity paradox" emerges as obesity correlates with better clinical outcomes in patients with metastatic melanoma, especially when treated with anti PD-1. In a retrospective study of 266  patients with metastatic melanoma treated with pembrolizumab or nivolumab, it emerges that obesity had a protective role, specifically a 10 cm2/m2 increase in visceral fat index (VFI) was associated with longer overall survival after adjusting for covariates, but this was not maintained when adjusted for systemic immune-inflammation index. Thus, the prognostic role of visceral obesity is conditional on inflammatory status. Obesity appears to determine an increase in PD-1 expression, induces T-cells to release more PD-1 protein, and increases the secretion of adiponectin and leptin from adipose tissue. These alterations result in increased T-cell depletion and dysfunction, which promote tumour progression. On the other hand, this provides a better understanding of the relationship between obesity and ICIs, as these agents remove the inhibitory signals of T-cell activation and promote an effective anti-tumour response. [44]

It is important to notiche that therapy with pembrolizumab can be combined with BRAF inhibitors. The BRAF V600 mutation is found in approximately 35-50% of metastatic melanomas, and is associated with increased disease aggressiveness and reduced survival [69,70].  A recent multicenter, double-blind phase III trial (STARBOARD) of 624 patients with BRAF V600E/K metastatic melanoma compared the efficacy of triple therapy (encorafenib, binimetinib and pembrolizumab) versus placebo plus PD-1 monotherapy (pembrolizumab). However, data from this study are not currently available.

Finally, a recent meta-analysis by Li et al. [71] evaluated the efficacy and safety of ICIs in 10090 patients from randomised clinical trials. These data show that pembrolizumab at a dosage of 10 mg/kg is the treatment option with the better efficacy and safety, equally with nivolumab, among ICIs in monotherapy. However, the best OS and PFS are obtained by combining nivolumab 1 mg/kg and ipilimumab 3mg/kg.

  1. Conclusions

In conclusion, pembrolizumab represents a fundamental option in the therapy of metastatic melanoma, as confirmed by data in real-life and in long-term follow-up. However, a certain proportion of patients do not respond to this therapy and therefore new combination options need to be evaluated. Among the new treatments, the use of vaccinations tailored to tumor epitopes could represent a breakthrough in the treatment of resistant forms, but further studies with larger sample numbers are needed to confirm preliminary results. Biomarkers that may allow selection of patients with a better chance of response to pembrolizumab therapy are also very important. In particular, the evaluation of CD103 expression and CT characteristics of metastases could in the future be easily integrated into diagnostic practices, allowing a more specific therapeutic choice. However, further studies with a larger sample are required to confirm the validity of these biomarkers.

Reviewer 2 Report

thank you for thee  opportunity to review this manuscript. 

my suggestions:

Melanoma is one of the most problematic malignancies for public health, with a progressive increase in new diagnoses. – this sentence needs to be rewritten, avoid vague statements, what does problematic entail and what does the increase in new diagnosis mean? Higher number of people getting it, higher number of people diagnosed because of better screening, or higher number of new types of melanoma or something else?

 The structure of the review is misleading – it is called a narrative review, then exclusion and inclusion criteria are mentioned. If this is narrative review, these details on search strategy are unnecessary or if authors decide to keep this information, it should be described in more detail. What do you mean by articles not meeting the search criteria –  and why repeat the purpose in the material and methods? I suggest to just replace materials and methods and first part of the results with few sentences describing the need for this review, what does it add to other reviews on this topic.  More importantly, previous reviews are not mentioned, and there must be either mention of those already published or a statement that this is a first review of this kind. There must be a statement on what is the purpose of this review with respect to those already published, what will this paper add.

It seems that both clinical trials and observational studies were included, and this should be emphasized more.

Line 74 – “the” is not needed before patient

What does this title mean - Pembrolizumab and associations, what are associations?

In my opinion, a paragraph of registered trials, that are still happening is missing, to get the full scope of the topic.

 Is the discussion necessary as a separate part? I suggest to reorganize key points from the discussion in the text presented in the results. 

 I am unsure on the paragraph about obesity  - it is just one study, does it need to be a separate title

thank you for thee  opportunity to review this manuscript. 

my suggestions:

Melanoma is one of the most problematic malignancies for public health, with a progressive increase in new diagnoses. – this sentence needs to be rewritten, avoid vague statements, what does problematic entail and what does the increase in new diagnosis mean? Higher number of people getting it, higher number of people diagnosed because of better screening, or higher number of new types of melanoma or something else?

 The structure of the review is misleading – it is called a narrative review, then exclusion and inclusion criteria are mentioned. If this is narrative review, these details on search strategy are unnecessary or if authors decide to keep this information, it should be described in more detail. What do you mean by articles not meeting the search criteria –  and why repeat the purpose in the material and methods? I suggest to just replace materials and methods and first part of the results with few sentences describing the need for this review, what does it add to other reviews on this topic.  More importantly, previous reviews are not mentioned, and there must be either mention of those already published or a statement that this is a first review of this kind. There must be a statement on what is the purpose of this review with respect to those already published, what will this paper add.

It seems that both clinical trials and observational studies were included, and this should be emphasized more.

Line 74 – “the” is not needed before patient

What does this title mean - Pembrolizumab and associations, what are associations?

In my opinion, a paragraph of registered trials, that are still happening is missing, to get the full scope of the topic.

 Is the discussion necessary as a separate part? I suggest to reorganize key points from the discussion in the text presented in the results. 

 I am unsure on the paragraph about obesity  - it is just one study, does it need to be a separate title

Author Response

my suggestions:

Melanoma is one of the most problematic malignancies for public health, with a progressive increase in new diagnoses. – this sentence needs to be rewritten, avoid vague statements, what does problematic entail and what does the increase in new diagnosis mean? Higher number of people getting it, higher number of people diagnosed because of better screening, or higher number of new types of melanoma or something else?

Done

Melanoma is one of the most problematic malignancies for public health, since melanoma is the most life-threatening skin cancer, associated with 75 per cent of skin cancer deaths. The progressive increase in new diagnoses appears to be due to improvements in surveillance and early diagnosis of melanoma, although sun exposure and sunbeds also seem to play an important role.

The structure of the review is misleading – it is called a narrative review, then exclusion and inclusion criteria are mentioned. If this is narrative review, these details on search strategy are unnecessary or if authors decide to keep this information, it should be described in more detail. What do you mean by articles not meeting the search criteria –  and why repeat the purpose in the material and methods? I suggest to just replace materials and methods and first part of the results with few sentences describing the need for this review, what does it add to other reviews on this topic. 

In accordance with the journal's requirements for revisions, the material and methods section is required. We have therefore decided to expand this section as follows:

  1. Materials and Methods

A search was conducted on Pubmed using "pembrolizumab," and "metastatic melanoma" as keywords. Only English language studies were included in the narrative review. Studies from 2022 onward were considered, as a time limit. We identified a total of 162 studies. Only studies, including clinical trials, observational studies or metanalysis, reporting efficacy data about pembrolizumab, alone or in combination or in specific conditions, were included. Studies on safety or adverse effects were not included in this narrative review, as well as case-report or case-series. We decided to include a total of 26 studies that met the inclusion criteria. We reported studies, even with preliminary data, from clinical studies that were published from 2022 onwards, while ongoing clinical trials with no published data were not included in this review. The purpose is to summarize the latest evidence produced and reviewed regarding the efficacy of pembrolizumab in patients with metastatic melanoma, since, at the best of our knowledge, a monothematic and specific overview about pembrolizumab on the most recent studies is not currently available in the literature.

More importantly, previous reviews are not mentioned, and there must be either mention of those already published or a statement that this is a first review of this kind. There must be a statement on what is the purpose of this review with respect to those already published, what will this paper add.

The purpose is to summarize the latest evidence produced and reviewed regarding the efficacy of pembrolizumab in the patient with metastatic melanoma, since, at the best of our knowledge, a monothematic and specific overview about pembrolizumab on the most recent studies is not currently available in the literature.

It seems that both clinical trials and observational studies were included, and this should be emphasized more.

Done,

Only studies, including clinical trials, observational studies or metanalysis, reporting efficacy data about pembrolizumab, alone or in combination or in specific conditions, were included.

Line 74 – “the” is not needed before patient

Done, thank you.

What does this title mean - Pembrolizumab and associations, what are associations?

In my opinion, a paragraph of registered trials, that are still happening is missing, to get the full scope of the topic.

We reported studies, even with preliminary data, from clinical studies that were published from 2022 onwards, while ongoing clinical trials with no published data were not included in this review. The purpose is to summarize the latest evidence produced and reviewed regarding the efficacy of pembrolizumab in patients with metastatic melanoma.

 Is the discussion necessary as a separate part? I suggest to reorganize key points from the discussion in the text presented in the results. 

We decided to structure the review in this way as required by the journal guidelines. In addition, another reviewer asked us to expand the discussion and conclusion sections:

 I am unsure on the paragraph about obesity  - it is just one study, does it need to be a separate title

We removed the paragraph and added in discussion

 Furthermore, evaluation of the tumor microenvironment, appears to be useful in assessing the prognosis of melanoma [64-68], and to be able to give insight into the efficacy of pembrolizumab therapy [35], although the data need further confirmation on larger samples. For example, although obesity is a major risk factor for cancer mortality [43], an "obesity paradox" emerges as obesity correlates with better clinical outcomes in patients with metastatic melanoma, especially when treated with anti PD-1. In a retrospective study of 266  patients with metastatic melanoma treated with pembrolizumab or nivolumab, it emerges that obesity had a protective role, specifically a 10 cm2/m2 increase in visceral fat index (VFI) was associated with longer overall survival after adjusting for covariates, but this was not maintained when adjusted for systemic immune-inflammation index. Thus, the prognostic role of visceral obesity is conditional on inflammatory status. Obesity appears to determine an increase in PD-1 expression, induces T-cells to release more PD-1 protein, and increases the secretion of adiponectin and leptin from adipose tissue. These alterations result in increased T-cell depletion and dysfunction, which promote tumour progression. On the other hand, this provides a better understanding of the relationship between obesity and ICIs, as these agents remove the inhibitory signals of T-cell activation and promote an effective anti-tumour response. [44]

It is important to notiche that therapy with pembrolizumab can be combined with BRAF inhibitors. The BRAF V600 mutation is found in approximately 35-50% of metastatic melanomas, and is associated with increased disease aggressiveness and reduced survival [69,70].  A recent multicenter, double-blind phase III trial (STARBOARD) of 624 patients with BRAF V600E/K metastatic melanoma compared the efficacy of triple therapy (encorafenib, binimetinib and pembrolizumab) versus placebo plus PD-1 monotherapy (pembrolizumab). However, data from this study are not currently available.

Finally, a recent meta-analysis by Li et al. [71] evaluated the efficacy and safety of ICIs in 10090 patients from randomised clinical trials. These data show that pembrolizumab at a dosage of 10 mg/kg is the treatment option with the better efficacy and safety, equally with nivolumab, among ICIs in monotherapy. However, the best OS and PFS are obtained by combining nivolumab 1 mg/kg and ipilimumab 3mg/kg.

Round 2

Reviewer 1 Report

The authors have addressed my comments satisfactorily.

Author Response

We thank you for the valuable suggestions that have allowed us to improve our manuscript